# An infodemiologic review of internet resources on dental hypersensitivity: A quality and readability assessment

**Muath Saad Alassaf**[1]*, **Ayman Bakkari**[2], **Jehad Saleh**[2], **Abdulsamad Habeeb**[2], **Bashaer Fahad Aljuhani**[3], **Ahmad A. Qazali**[4], **Ahmed Yaseen Alqutaibi**[4,5]

1 Department of Oral and Maxillofacial Surgery, King Fahad Hospital, Madina, Saudi Arabia, 2 College of Dentistry, Taibah University, Medina, Saudi Arabia, 3 The Bright Smile Horizon Dental Clinic, Medina, Saudi Arabia, 4 Substitutive Dental Sciences Department (Prosthodontics), College of Dentistry, Taibah University, Al Madinah, Saudi Arabia, 5 Department of Prosthodontics, College of Dentistry, Ibb University, Ibb, Yemen

* Mo3ath345@gmail.com

## Abstract

### Background

This study aimed to investigate the quality and readability of online English health information about dental sensitivity and how patients evaluate and utilize these web-based information.

### Methods

The credibility and readability of health information was obtained from three search engines. We conducted searches in "incognito" mode to reduce the possibility of biases. Quality assessment utilized JAMA benchmarks, the DISCERN tool, and HONcode. Readability was analyzed using the SMOG, FRE, and FKGL indices.

### Results

Out of 600 websites, 90 were included, with 62.2% affiliated with dental or medical centers, among these websites, 80% exclusively related to dental implant treatments. Regarding JAMA benchmarks, currency was the most commonly achieved and 87.8% of websites fell into the "moderate quality" category. Word and sentence counts ranged widely with a mean of 815.7 (±435.4) and 60.2 (±33.3), respectively. FKGL averaging 8.6 (±1.6), SMOG scores averaging 7.6 (±1.1), and FRE scale showed a mean of 58.28 (±9.1), with "fair difficult" being the most common category.

### Conclusion

The overall evaluation using DISCERN indicated a moderate quality level, with a notable absence of referencing. JAMA benchmarks revealed a general non-adherence among websites, as none of the websites met all of the four criteria. Only one website was HON code certified, suggesting a lack of reliable sources for web-based health information accuracy. Readability assessments showed varying results, with the majority being "fair difficult".

**Data Availability Statement:** All relevant data is within the manuscript and its Supporting Information files are uploaded.

**Funding:** The author(s) received no specific funding for this work.

**Competing interests:** The authors have declared that no competing interests exist.

Although readability did not significantly differ across affiliations, a wide range of the number of words and sentences count was observed between them.

## Introduction

Patient health information seeking has become increasingly common in today's digital age. Patients often seek online health information to gain a better understanding of their dental conditions to make informed decisions about their healthcare options. However, self-diagnosis through the internet may lead to misinterpretation of symptoms and the selection of incorrect treatment options, such as using toxic herbs, unlicensed remedies, or inaccurate prophylactic strategies [1]. Therefore, evaluating the quality of information published on these websites is crucial. Information technology is revolutionizing the traditional healthcare paradigm by reorienting emphasis from treatment to prevention. As per the World Health Organization, a noteworthy 71% of individuals utilizing the internet actively seek health-related information online [1]. In the field of dentistry, patients commonly seek information regarding dental conditions, such as teeth hypersensitivity. This entails gathering knowledge about their specific condition, available treatment options, and potential causative factors, such as the exposure of open dentinal tubules due to stimuli like cold beverages or cold air [2–4]. While clinical investigations have shed light on the prevalence of dental pain, there is still a lack of understanding regarding the underlying processes that contribute to tooth discomfort [5]. The structure of dentin, specifically dentinal tubules, plays a vital role in tooth sensitivity [6]. Furthermore, research suggests that there are gender-based disparities, with women typically reporting higher levels of pain and lower pain thresholds compared to men [7].

Several hypotheses contribute to the understanding of dental pain in an academic context. These include the role of temperature-sensitive channels in dentin, the involvement of odontoblasts in the sensation of pressures, and the hydrodynamic theory suggesting that pain arises from fluid movement within dentinal tubules [8]. This particular type of pain, characterized by sharp sensations, is expected to become more prevalent as dental tissue longevity increases and tooth wear occurs more frequently [9]. The hydrodynamic theory elucidates the mechanism through which pain is initiated. Specifically, stimuli such as exposure to cold, air blasts, and variations in pressure lead to swift fluid displacement within the dentinal tubules, thereby stimulating nerve fibers in close proximity to the pulp-dentine interface [10]. Consequently, dentists are tasked with the important responsibility of accurately diagnosing and effectively managing various oral pathologies that may present similar symptoms to teeth hypersensitivity. These oral pathologies include postoperative sensitivity, dental caries, cracked teeth, and dental leakage [11]. The primary cause of dentin hypersensitivity often involves gingival recession, which exposes the cementum and dentinal tubules to external stimuli. Factors that worsen this condition include aggressive toothbrushing, poor oral hygiene, abrasive toothpaste, periodontal diseases, facial piercings, orthodontic treatments, and inherent susceptibility [12, 13]. Interestingly, studies have yielded conflicting results regarding the relationship between plaque buildup and hypersensitivity, highlighting the complexity of these interactions [14, 15]. Additionally, there is a significant link between facial piercings and increased gingival recession, emphasizing the need for patient education regarding potential consequences [16]. Orthodontic treatments can also contribute to gingival recession and subsequent sensitivity [17]. To manage dentinal hypersensitivity, toothpastes often incorporate ingredients like potassium nitrate to desensitize nerves and strontium to occlude dentinal tubules with crystal

formations, while fluoride aids in remineralizing tooth surfaces to alleviate sensitivity [18]. These strategies aim to mitigate pain by addressing the underlying mechanisms identified through the hydrodynamic theory. Research suggests that individuals who actively search for health information on the internet tend to have greater health concerns. In particular, adults who engage in these activities often perceive their health as being worse off compared to those who do not seek health information online. Likewise, young people who use online resources to search for health-related information are more likely to identify clinical issues or symptoms of depression in themselves, unlike their peers who do not use online resources [19]. Due to the widespread availability of smartphones, tablets, and laptops, a considerable percentage of individuals, estimated at approximately 52%, have resorted to utilizing the internet as a source for health-related information and symptoms in 2022. Notably, this upward trend in dependence on digital platforms for health-related information is expected to endure [20].

Our study sought to evaluate and analyze the quality and readability of online English health information pertaining to dental sensitivity. Additionally, we aimed to explore how patients can effectively evaluate such information, the implications thereof, and the significance of awareness in an era characterized by easy access to information.

## Materials and methods

In this Endemiological Review, our dataset was sourced from publicly available websites. Data collection was conducted through manual search engines and the collected data was recorded in a spreadsheet. The analysis was performed using SPSS. All processes strictly adhered to the terms and conditions specified by the data provider.

### Search strategy

Using Google Chrome, version 81.0.4044, we conducted searches on Yahoo! (http://www.yahoo.com), Bing (http://www.bing.com), and Google (http://www.google.com). This approach is in line with the findings of "The Pew Research Center's Internet and American Life Project" [21], which reports that 79% of individuals who search for health information online utilize these search engines. Before commencing our searches, we cleared all cookies to minimize any potential impact from past search activities. To eliminate any bias from search history, we performed our browsing in "incognito" mode. We identified and eliminated any duplicate website listings. We collected pertinent English-language websites that offer health information on tooth sensitivity for further evaluation. Website exclusion criteria encompassed: (1) content presented in languages other than English; (2) information conveyed exclusively through tips or solely in video or audio formats; (3) complete textbooks or full scientific papers; (4) websites featuring sponsored links, banner ads, or discussion forums; (5) sites that were not directly accessible or were blocked, necessitating identification and password for entry; and (6) social media platforms and forums. Websites that did not fall into these exclusion categories were retained for subsequent analysis of quality and readability.

The websites included in this study were categorized according to their affiliation, which encompassed four categories: commercial, non-profit organization, dental/medical center, and Governmental/University. Subsequently, the websites were classified as either exclusively related (if they were solely dedicated to dental health) or partially related. The evaluation of the websites also took into account the type of content and its presentation. Quality assessment tools

To evaluate the quality of the chosen websites, we employed several instruments: the JAMA benchmarks [22], the DISCERN instrument [22], and the Health on the Net Foundation Code of Conduct (HONcode) [23]. The HONcode tool is capable of authorizing the display of a seal

(HON award-like badge) on a specific website, provided that the website meets the HONcode standards. This seal is valid for a period of one year and serves as a form of certification badge.

The benchmarking tool provided by the Journal of the American Medical Association evaluates various key aspects: authorship (including information about the authors, their contributions, qualifications, and institutional affiliations); attribution (the presence of clear sources and references for cited content); disclosure (information regarding ownership, advertisements, sponsorships, commercial support, or any potential conflicts of interest); and timeliness (explicit mention of the dates for the initial publication and subsequent updates of the material). Websites are awarded one point for each criterion met (indicated by a "yes" response); if a criterion is not met, zero points are given. The total possible score from JAMA ranges from 0 (no criteria satisfied) to 4 (all criteria satisfied). The DISCERN instrument consists of 16 questions divided into three sections: questions 1–8 aim to assess the reliability of websites as sources of information on specific treatments; questions 9–15 evaluate the range of treatment options presented; and question 16 appraises the overall quality rating. Each question is rated on a scale from 1 to 5, where a score of 1 indicates a substandard website and 5 indicates a high-quality website.

Two researchers conducted a comprehensive assessment of websites using the JAMA and DISCERN tools. Initially, each researcher individually evaluated five websites, and any disagreements in their evaluations were resolved through discussion. Subsequently, inter-rater reliability was calculated for all assessed websites. Moreover, the researchers installed the HONcode software as a browser extension in Google Chrome. This software highlighted websites that were certified by HONcode by displaying a HONcode seal during searches. To ensure the validity of the certification, websites displaying the HONcode seal were cross-checked with the main HONcode website to confirm that their certification was current.

## Readability assessment

For readability, an online readability calculator tool was used on all of the websites that were assessed [24]. This tool has been specifically designed for the evaluation of English texts. However, it can also be utilized for other languages. The website employs widely recognized and commonly used analytical tools to assess the text, including the Flesch Kincaid grade level (FKGL), Flesch reading ease (FRE), and Simple Measure of Gobbledygook (SMOG). These readability analysis tools, namely SMOG, FRE, and FKGL, have been carefully chosen for the evaluation of the texts. The FRES is derived by employing a specific mathematical formula that takes into account both the average sentence length and the number of syllables per word. The resulting scores range from 0 to 100, with higher scores denoting a higher degree of readability. Texts with scores above 90 are generally comprehensible for 5th-grade students, while those falling between 60 and 70 are deemed suitable for students in 8th and 9th grades. Scores below 50 indicate that the text possesses an elevated level of academic complexity. FKGL incorporates the tenets of the FRES by incorporating variables such as words per sentence and syllables per word. On the other hand, the SMOG index assesses the number of polysyllabic words and estimates the level of education needed to grasp the text. Both FKGL and SMOG yield a grade level indicator for comprehension, where a score like 7.4 implies that a student at a seventh-grade level should be able to comprehend the text. Nevertheless, for enhanced comprehension, it is recommended that FKGL and SMOG scores remain at 5 or below [25].

## Statistical analysis

All statistical analyses were conducted using SPSS software, version 21.0 (IBM Corp., Armonk, NY). The data were presented either as percentages and frequencies, or as means with standard

deviations (SDs), depending on their characteristics. To examine differences among various website categories for quantitative variables, the Kruskal-Wallis test was applied, with the Bonferroni correction being used for pairwise comparisons. The Kruskal-Wallis test is a non-parametric statistical test utilized to assess whether there are significant differences between the medians of three or more independent groups. This test serves as an alternative to the one-way ANOVA when the assumptions of normality are not satisfied. Fisher's exact test or the Chi-square test, as appropriate, were utilized to explore potential relationships between website categories and qualitative variables. Statistical significance was determined at a P value of less than 0.05.

## Results

### Available websites and categorization

A search conducted on Google, Bing, and Yahoo using the keywords "Sensitive teeth" and "Teeth hypersensitivity" generated more than 30 million results. Following the application of eligibility criteria, the first 100 websites for each search term from each search engine were assessed. Out of the initial sample of 600 websites, 90 websites satisfied the inclusion requirements. The exclusion process is illustrated in Fig 1. In terms of affiliation, 56 websites (62.2%) were associated with dental or medical centers, while 14 websites (15.6%) were commercial entities, 13 websites (14.4%) were non-profit organizations, and 7 websites (7.8%) were linked to government bodies or universities. The majority of the included websites, 72 (80%), provided information specifically on dental implant treatments. None of the evaluated websites contained personal anecdotes or audio content. A summary of the content types and website categories is presented in Table 1.

### Quality assessment

Quality evaluations were conducted using the DISCERN, JAMA benchmarks, and the HON seal. The average DISCERN score was 41.23 (±7.63). The average scores for the sections on reliability and treatment were 19.48 (±3.68) and 19.11 (±4.58), respectively. Table 2 presents the average scores for each DISCERN question. The lowest quality score was associated with the fourth question, which pertained to the clarity of the sources used in compiling the publication, while the highest score was linked to the first question regarding the clarity of objectives. Overall, the quality scores classified the majority of the websites, 79 (87.8%), as having moderate quality. Only eleven websites were categorized as having low quality. Table 3 illustrates the distribution of quality scores based on the affiliations of the websites. No significant correlation was found between the DISCERN quality categories and website affiliation. However, as shown in Table 4, the average score for treatment quality was statistically significant when comparing commercial websites to those affiliated with dental or medical centers, with a p-value of less than 0.05. In relation to the JAMA benchmarks, none of the websites fulfilled all four criteria. However, four websites satisfied three of the benchmarks, two of which were affiliated with non-profit organizations. There was no significant difference in the number of benchmarks fulfilled among different website affiliations, as indicated by a p-value of 0.87. The JAMA benchmarks encompass four criteria, with currency being the most frequently met, observed on 35 websites, followed by attribution, which was present on nine sites. Only eight websites displayed disclosure. A comprehensive breakdown of the performance of different affiliations in the JAMA evaluation is provided in Table 3. As for the HON code certification, only the Mayo Clinic website (www.mayoclinic.org) received accreditation.

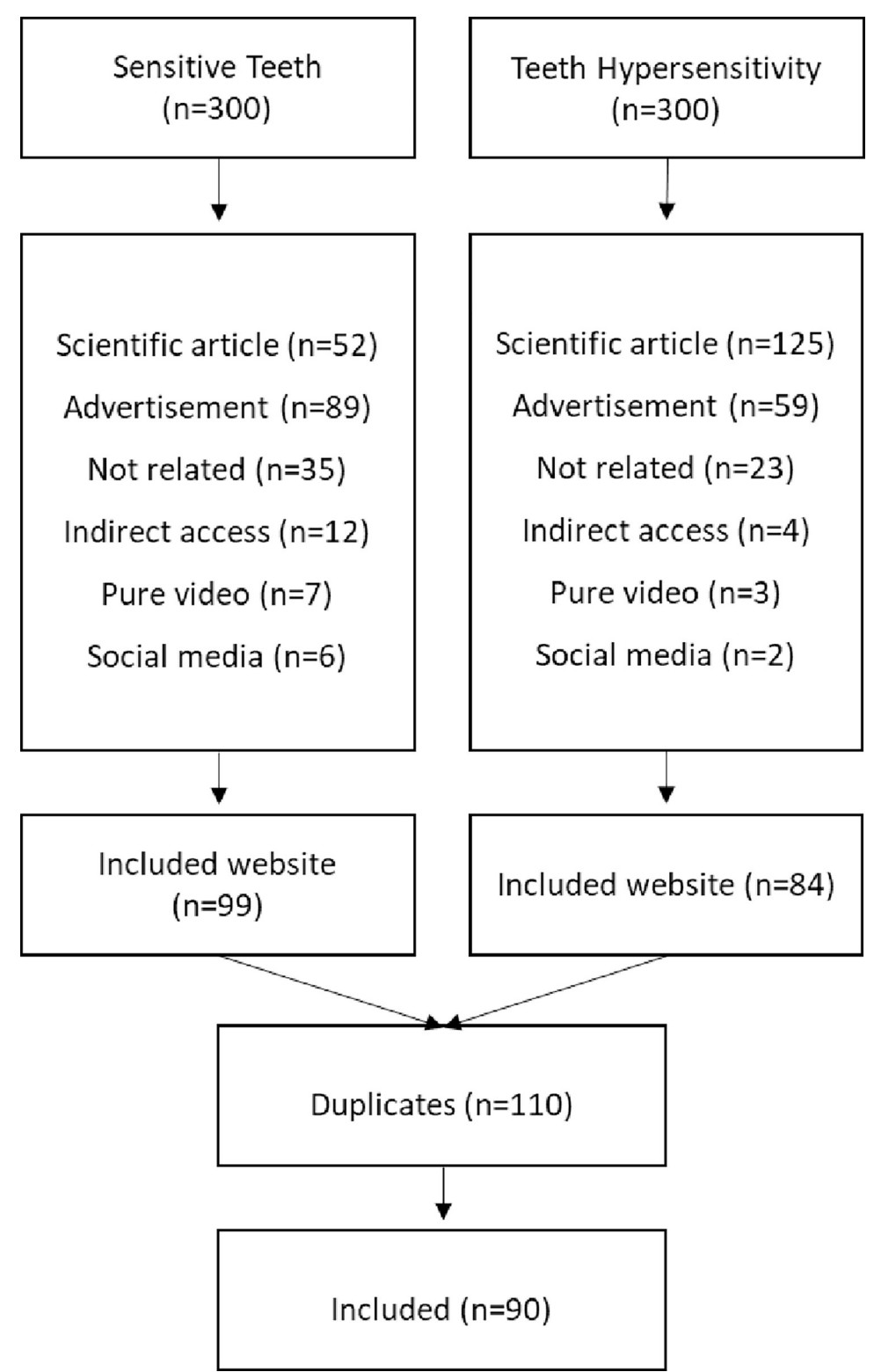

**Fig 1. Flow chart of the searching strategy.**

**Table 1. Categorization of websites based on affiliation, specialization, content type, and content presentation (n = 90).**

| Category | Criteria | Number of websites | Percentage |
|---|---|---|---|
| Affiliation | Commercial | 14 | 15.6 |
| | Non-profit organization | 13 | 14.4 |
| | Dental/medical center | 56 | 62.2 |
| | Governmental/University | 7 | 7.8 |
| Specialization | Exclusively related | 72 | 80 |
| | Partly related | 18 | 20 |
| Content type | Medical facts | 89 | 98.9 |
| | Clinical trials | 1 | 1.1 |
| | Human interest stories | 0 | 0 |
| | Question and answer | 63 | 70 |
| Content presentation | Image | 25 | 27.8 |
| | Video | 4 | 4.4 |
| | Audio | 0 | 0 |

## Readability

The scores of the FRE ranged from 40 to 84, with an average score of 58.28 (±9.1). The most common level of readability was categorized as "fairly difficult," encompassing 34 websites, while 31 sites were considered to have a "standard" level of readability. Only 8 websites were classified as "fairly easy." Fig 2 illustrates the distribution of FRE categories based on the websites' affiliations. The KGL scores varied from 3.7 to 13.5, with an average of 8.6 (±1.6). The SMOG index scores ranged from 4.7 to 10.6, with an average of 7.6 (±1.1). As shown in Table 4, there were no statistically significant differences in average readability scores among the different affiliations. The number of words and sentences varied across the websites, ranging from 96 to 1995 with a mean of 815.7 (±435.4) and 8 to 167 with a mean of 60.2 (±33.3), respectively. A statistically significant difference was observed among the affiliations, as presented in Table 4.

**Table 2. Means and standard deviation scores for DISCERN instrument (n = 90).**

| Domain | DISCERN question | Mean (SD) | Max-Min |
|---|---|---|---|
| RELIABILITY | Q1. Explicit aims | 3.93 (0.64) | 5.0–2.0 |
| | Q2. Aims achieved | 3.44 (0.78) | 5.0–1.0 |
| | Q3. Relevance | 3.57 (0.68) | 5.0–2.0 |
| | Q4. Explicit sources | 1.27 (0.87) | 5.0–1.0 |
| | Q5. Explicit date | 2.31 (1.64) | 5.0–1.0 |
| | Q6. Balanced and unbiased | 2.84 (0.99) | 5.0–1.0 |
| | Q7. Additional sources | 2.12 (0.94) | 5.0–1.0 |
| | Q8. Areas of uncertainty | 2.68 (0.73) | 4.0–1.0 |
| TREATMENT OPTIONS | Q9. How treatment works | 2.98 (1.07) | 5.0–1.0 |
| | Q10. Benefits of treatment | 2.76 (1.00) | 4.0–1.0 |
| | Q11. Risk of treatment | 1.39 (0.70) | 4.0–1.0 |
| | Q12. Effects of no treatment | 1.34 (0.65) | 3.5–1.0 |
| | Q13. Effects on quality of life | 2.72 (0.94) | 5.0–1.0 |
| | Q14. All alternatives described | 3.14 (1.16) | 5.0–1.0 |
| | Q15. Shared decision | 2.06 (0.88) | 4.0–1.0 |
| OVERALL RATING | | 2.64 (0.56) | 4.0–1.0 |

**Table 3. Quality and readability of the included websites based on their affiliation reported as frequency and percentage (n = 90).**

| Variable | Variable type | Commercial | Dental/Medical center | Governmental/ University | Non-profit organization | Total | P-value |
|---|---|---|---|---|---|---|---|
| Number of achieved JAMA items per website | None | 10 | 32 | 1 | 5 | 48 | 0.087 |
| | One | 3 | 19 | 5 | 4 | 31 | |
| | Two | 1 | 4 | 0 | 2 | 7 | |
| | Three | 0 | 1 | 1 | 2 | 4 | |
| | Four | 0 | 0 | 0 | 0 | 0 | |
| JAMA items | Authorship | 1 | 1 | 1 | 2 | 5 | 0.172 |
| | Attribution | 0 | 4 | 0 | 5 | 9 | 0.002* |
| | Disclosure | 1 | 5 | 1 | 1 | 8 | 0.955 |
| | Currency | 3 | 20 | 6 | 6 | 35 | 0.034* |
| DISCERN | Low | 4 | 5 | 0 | 2 | 11 | 0.161 |
| | Medium | 10 | 51 | 7 | 11 | 79 | |
| | High | 0 | 0 | 0 | 0 | 0 | |

## Discussion

Several terms have been employed in the academic literature to elucidate the phenomenon of dentinal hypersensitivity. These include dentin sensitivity, cervical hypersensitivity, cemental hypersensitivity, and root hypersensitivity. Each of these terms corresponds to the particular site where sensitivity manifests, such as the cervical, root, dentin, and cemental structures. It should be noted that all of these terms are synonymous and may be used interchangeably to denote the identical clinical concept [25]. The presence of such a condition can have a negative impact on the quality of life. Everyday activities like eating, drinking, speaking, and tooth-brushing may frequently be accompanied by discomfort [26].

A recent systematic review conducted in 2019 revealed that the estimated prevalence of dentin hypersensitivity was approximately 11.5%. The review noted a significant degree of heterogeneity among the included studies, which could only be partially accounted for by variations in study characteristics. Consequently, it was suggested that a new prevalence study could potentially yield a wide range of rates, ranging from 4.8% to 62.3% [27]. However, the exact prevalence is still a topic of debate, as earlier research suggests a significantly higher

**Table 4. Comparison between means according to websites' affiliation (n = 90).**

| Variable | Commercial | Dental/Medical center | Governmental/University | Non-profit organization | P-value |
|---|---|---|---|---|---|
| DISCERN Scores | | | | | |
| Overall | 36.4 (7.5) | 42.4 (7.3) | 40.5 (4.4) | 41.9 (9.2) | 0.081 |
| Reliability | 18.5 (3.7) | 19.4 (3.1) | 19.5 (4.4) | 20.8 (5.3) | 0.851 |
| Treatment | 15.6 (4.0) | 20.2 (4.4) | 18.1 (3.4) | 18.5 (4.8) | 0.006* |
| Readability Formulas | | | | | |
| FRES | 55.3 (11.8) | 59.1 (8.1) | 56.7 (9.6) | 59.0 (10.1) | 0.473 |
| FKGL | 8.8 (1.8) | 8.6 (1.6) | 8.8 (1.7) | 8.6 (1.7) | 0.635 |
| SMOG | 7.3 (1.0) | 7.6 (1.1) | 7.4 (0.9) | 7.8 (0.9) | 0.790 |
| Words | 497.9 (277.0) | 891.6 (417.0) | 690.6 (335.3) | 898.5 (555.9) | 0.007** |
| Sentences | 38.4 (21.1) | 65.8 (32.4) | 50.7 (24.4) | 65.0 (43.2) | 0.015*** |

* Between Commercial and Dental/medical centers, p-value = 0.001

** Between Commercial and Non-profit organizations, p-value = 0.028; also, between Commercial and Dental/medical centers, p-value = 0.001

*** Between Commercial and Non-profit organizations, p-value = 0.045; also, between Commercial and Dental/medical centers, p-value = 0.002

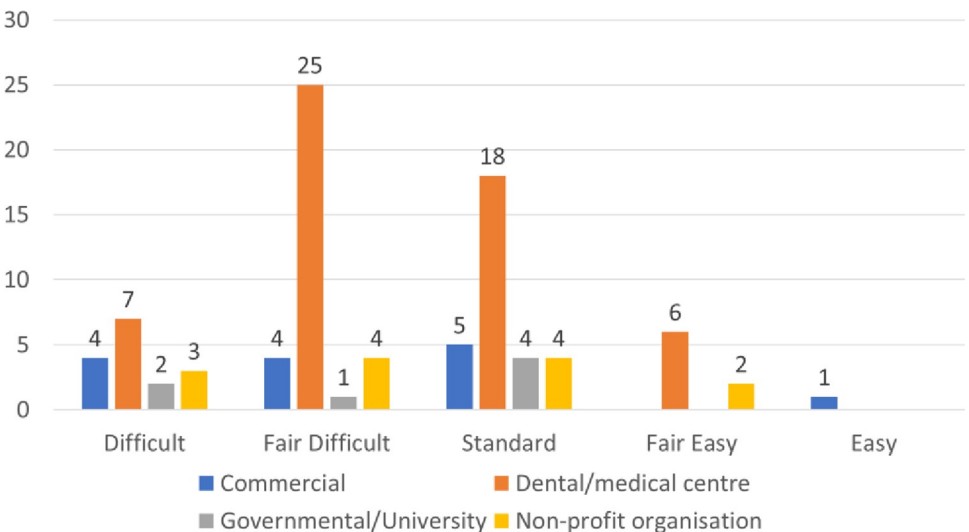

**Fig 2. The FRES difficulty categories of the evaluated websites (n = 90).**

prevalence [28, 29]. Despite the accuracy of the current hypersensitivity estimate, it undoubt-edly encourages individuals to seek treatment or find information about their condition. Given that sensitivity typically is not urgent, individuals might tend to seek online health infor-mation to rule out potential issues or seek advice from a dentist [30]. This inclination stems from people's tendency to embrace positive self-assessments, like indications of good health, over negative ones, such as signs of illness [31]. This aspect is crucial as it encourages patients to adopt a more proactive approach in managing their healthcare [32]. The objective of this study was to assess and analyze the quality and readability of English health information avail-able online regarding dental hypersensitivity. Additionally, the study aimed to examine how patients evaluate this information, the implications of such assessment, and the significance of public awareness in the age of easily accessible information and knowledge. Three search engines were utilized to gather the data: Google, Bing, and Yahoo. Together, these search engines provided access to over 30,000,000 websites. The first 100 websites from each search engine were reviewed to determine inclusion or exclusion. The exclusion criteria were limited to websites containing scientific articles, advertisements, no direct website access, pure video content, social media platforms, and languages other than English. The most commonly excluded sites were scientific articles related to "teeth hypersensitivity," as this term is predomi-nantly used in a medical context. Conversely, when searching for "sensitive teeth," advertise-ment websites accounted for the highest number of excluded samples. Out of all the websites reviewed, only 90 met the inclusion criteria.

Categorization involved dividing websites based on their affiliation, such as commercial, non-profit organization, medical/dental center, or governmental/university related, as well as whether they were totally or partially related to hypersensitivity. The majority of information related to our research topic was found on dental or medical centers' websites, which included 56 webpages, surpassing the combined number of all other categories. This may be due to hos-pitals and clinics indirectly advertising their services by providing background information on the treatments they offer. Governmental or university-based websites constituted the smallest category in terms of affiliation. This is a drawback in terms of the credibility of the informa-tion, as the American Medical Library Association emphasizes the importance of evaluating the source of information, whether it is from a governmental agency, academic institution,

research-affiliated organization, or commercial entity [33, 34]. What is worth noting about online information regarding hypersensitivity is its exclusivity. Unlike other researches where specific medical topics are only a portion of the website, hypersensitivity issues received significant interest, with 80% of the websites exclusively dedicated to this condition [35]. Most content was presented in the form of images. However, human interest stories and audios were not included in any of the website presentations, despite being suggested as highly important for online viewers [36, 37].

Regarding quality assessment, various tools were used, such as DISCERN, JAMA benchmark, and HON code. These three tools were mentioned and recommended in multiple studies [38–40]. The HONcode, DISCERN, and JAMA Benchmarks are among the most widely utilized instruments for evaluating the reliability and quality of information presented on medical websites. In comparison to other studies that used DISCERN as an assessment tool, the average score was relatively higher at 41.23 (±7.63) [41–43]. As the topic of hypersensitivity has numerous dedicated websites, this could lead to more detailed and comprehensive information, subsequently increasing the DISCERN and other tools' scores. The lowest DISCERN score was for question 4, which assessed the inclusion of references. Failing to provide valid references is not the ideal approach to presenting information and may lead to misunderstandings among patients, potentially resulting in the improper use of certain treatments [44]. However, it was suggested that certain populations selectively choose the information they reference and receive, which could make the absence of references on websites an unfavorable source for some individuals [45, 46]. Clear objectives were the most prevalent criteria presented on all websites, providing readers with a concise overview of the topic. The average provision of additional resources was 2.12 (0.94), which can be seen as a strong point for some websites. These additional resources, often referred to as referral links, improve the quality of the website and offer viewers a wider range of information [47].

The JAMA benchmarks tool, published by the Journal of the American Medical Association, evaluates four aspects: authorship, attribution, disclosure, and currency. The final JAMA score should range from 0 to 4, depending on the number of criteria achieved. Unfortunately, no website fulfilled all of these criteria, as seen in other studies [48, 49]. Almost half of the websites (48 websites) did not include any of the criteria. The highest score achieved was 3 out of 4, found on four websites. Currency was the most common achievement, present on 35 websites, followed by attribution on nine websites. Only eight websites had disclosure. The high score for the inclusion of dates on websites could be due to an auto-generated feature on some websites.

The HON code tool allows qualified websites to display a badge similar to the HON award. This badge, which is valid for one year, serves as a certificate-like symbol. The requirement to obtain this badge is compliance with the HON code criteria [50]. Out of all the websites included in this research, only one, Mayo Clinic, met the criteria and received approval. Considering the limited number of websites included in this study, it is not surprising that most websites found on Google are unlikely to be HON-certified [50]. It is important to note that there is a linguistic difference in the popularity of the HON code [51], which provides multilingual individuals with access to higher-quality online content.

The readability of a text is assessed using various readability indices that consider factors like the number of syllables or characters in the text. These indices measure the complexity of the vocabulary and sentence structure in written content and can be equated to a grade level, indicating the number of years of education required to understand the text [52]. The majority of the tools used for analysis are widely recognized as the leading readability assessment tools [53–55]. The results of the readability assessment, as determined by the Flesch Reading Ease (FRE) scale, indicate significant variation in the readability of the websites assessed. The FRE

scale ranged from 40 to 84, with an average of 58.28 (±9.1). The most common category was "fair difficult," which accounted for 34 websites, followed closely by "standard" in 31 websites. Only a small number of websites, specifically 8, were classified as "fair easy". Even websites within the same affiliation, such as commercial sites, exhibited a wide range of scores. This suggests that there is a considerable diversity in the complexity of language and sentence structures across websites and that categories do not necessarily correspond to the recommended level of education for readers. The Flesch-Kincaid Grade Level (FKGL) and the SMOG Index further highlight the variation in language complexity. FKGL ranged from 3.7 to 13.5, while the SMOG Index had a minimum score of 4.7 and a maximum score of 10.6. These readings underscore the heterogeneity in language complexity observed in online health content.

The analysis of word and sentence counts reveals significant variations in content length among websites. The word count ranged from 96 to 1995, while the sentence count ranged from 8 to 167. Content length varies according to the source. Commercial websites had the fewest words and sentences, possibly influenced by their focus on the potential benefits from website visitors. In contrast, medical or dental websites featured the highest word and sentence counts, nearly double that of commercial sites. However, this might cause online viewers to become bored or lose interest in reading the entire page, especially if they do not fully understand the topic [56]. To improve the status of websites providing dental health information, we suggest the following: Enhancing quality and reliability by striving to meet all JAMA benchmarks, including clear authorship, accurate attribution, full disclosure, and up-to-date information. Improving readability by writing content at a reading level accessible to the general public, ideally between the 5th and 8th-grade reading levels, as indicated by the Flesch-Kincaid Grade Level and SMOG index. Increasing certification by seeking approval from reputable sources like HONcode to assure users of their reliability and adherence to ethical standards. Expanding content variety by incorporating diverse content types, such as videos, human interest stories, and interactive tools, can engage users more effectively and cater to different learning preferences. Health practitioners, managers, and policymakers play a crucial role in ensuring the quality of online health information. We propose the following actions: Health practitioners should guide patients towards reputable sources and educate them on evaluating online information critically. Practitioners can also contribute content to reliable websites to enhance the quality and trustworthiness of available information. Managers should oversee the creation and maintenance of health websites, ensuring that content is accurate, up-to-date, and user-friendly. Implementing continuous quality improvement processes can help maintain high standards. Policy makers should develop and enforce regulations that mandate transparency, accuracy, and ethical standards for online health information. Supporting initiatives like HONcode certification can also promote higher quality standards across health websites.

## Limitations

Although the investigation included information from commonly used search engines such as Yahoo!, Bing, and Google, these platforms may not fully represent all the available online information on teeth sensitivity, as there are other free search engines available. In future studies, diversifying information sources beyond those sharing the same index, such as Bing and Yahoo, can enhance the breadth of research findings. By incorporating varying search engines, readers gain access to a wider array of perspectives and data, allowing for more comprehensive and varied insights. This approach not only enriches the research but also aids readers in selecting search engines that better meet their informational needs. Besides, both Bing and Yahoo share the same indexing and a significant portion of their technical back end. As a consequence, they produce search results that are similar in nature, although not completely

identical. Future research could explore the effectiveness and coverage of other search engines like Baidu, especially in regions with different internet regulations. By comparing search engines with distinct user bases, such as Baidu, Yahoo, Bing, and Google, researchers can gain insights into how users access English-language health information across diverse geographic and cultural contexts. This could help identify gaps in information availability and inform strategies to improve online health information dissemination globally. Additionally, examining user behavior and preferences on these platforms could provide a deeper understanding of search engine optimization and its impact on health literacy.

Furthermore, by excluding pure video and audio content, as well as social media, a significant portion of internet data was ignored. The inclusion of only the first 100 pages may be a potential limitation, given the dynamic nature of the internet. Due to the continuous uploading of information and ever-changing platform algorithms, a specific time frame and periodic checks are necessary as the order of website links can vary. Additionally, the study focused solely on English websites, which may restrict the sample size, and the selected keywords such as *Sensitive teeth* and *Teeth hypersensitivity* that may not accurately represent the terms patients use to describe their condition. Lastly, it should also be noted that Incognito mode does not prevent the transmission of IP address information to the search tool. As a result, search results can still be influenced by data that can be obtained from the IP address, such as geographic location.

## Conclusion

The analysis of online health information found that the majority of websites examined were affiliated with dental or medical centers, primarily focusing on dental implant treatments. However, the overall quality assessment using DISCERN criteria indicated a moderate level of quality, with a concerning lack of proper referencing across most sites. Moreover, adherence to JAMA benchmarks was notably deficient among all websites, with none meeting all four criteria, and only one site being HON code certified, highlighting significant gaps in reliability and accuracy of web-based health information. Moving forward, it is imperative for future research to address the shortcomings identified in the current literature on tooth sensitivity. Specifically, there is a need for studies that rigorously evaluate the quality and reliability of online health information pertaining to dental sensitivity. This includes exploring methodologies to enhance referencing practices, improve adherence to established quality benchmarks like those set by JAMA, and promote certifications such as the HON code for ensuring accurate and trustworthy online content. Furthermore, readability assessments revealed variability among websites, suggesting the importance of developing accessible and comprehensible health information for diverse patient populations. Future research should aim to standardize readability guidelines and explore innovative strategies to enhance the clarity and accessibility of dental health information online. This study highlights the moderate quality and challenging readability of online health information on dental sensitivity. The findings underscore the need for improved reference standards and HONcode certification to enhance reliability. By identifying gaps in current online resources, this research emphasizes the importance of developing accessible and trustworthy health information for patients.

## Supporting information

**S1 Appendix. List of the included websites.**
(DOCX)

**S1 Dataset.**
(XLSX)

## Author Contributions

**Conceptualization:** Muath Saad Alassaf, Ayman Bakkari, Jehad Saleh, Abdulsamad Habeeb, Bashaer Fahad Aljuhani, Ahmed Yaseen Alqutaibi.

**Data curation:** Muath Saad Alassaf, Ayman Bakkari, Jehad Saleh, Bashaer Fahad Aljuhani.

**Formal analysis:** Muath Saad Alassaf, Bashaer Fahad Aljuhani, Ahmed Yaseen Alqutaibi.

**Investigation:** Ayman Bakkari, Abdulsamad Habeeb, Bashaer Fahad Aljuhani.

**Methodology:** Muath Saad Alassaf, Ayman Bakkari, Abdulsamad Habeeb, Bashaer Fahad Aljuhani, Ahmad A. Qazali, Ahmed Yaseen Alqutaibi.

**Project administration:** Muath Saad Alassaf, Ahmed Yaseen Alqutaibi.

**Supervision:** Muath Saad Alassaf, Ahmad A. Qazali, Ahmed Yaseen Alqutaibi.

**Validation:** Ahmad A. Qazali.

**Visualization:** Ahmad A. Qazali.

**Writing – original draft:** Jehad Saleh, Abdulsamad Habeeb, Ahmad A. Qazali, Ahmed Yaseen Alqutaibi.

**Writing – review & editing:** Jehad Saleh, Abdulsamad Habeeb, Ahmad A. Qazali, Ahmed Yaseen Alqutaibi.

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
