## [Decision Letter · Decision Letter 0]

23 Jun 2024

PONE-D-24-10064An Infodemiologic Review of Internet Resources on Dental Hypersensitivity: A Quality and Readability Assessment.PLOS ONE

Dear Dr. Alassaf,

Thank you for submitting your manuscript to PLOS ONE. After careful consideration, we feel that it has merit but does not fully meet PLOS ONE’s publication criteria as it currently stands. Therefore, we invite you to submit a revised version of the manuscript that addresses the points raised during the review process.

Please submit your revised manuscript by Aug 07 2024 11:59PM. If you will need more time than this to complete your revisions, please reply to this message or contact the journal office at plosone@plos.org. Please include the following items when submitting your revised manuscript:A rebuttal letter that responds to each point raised by the academic editor and reviewer(s). You should upload this letter as a separate file labeled 'Response to Reviewers'.A marked-up copy of your manuscript that highlights changes made to the original version. You should upload this as a separate file labeled 'Revised Manuscript with Track Changes'.An unmarked version of your revised paper without tracked changes. You should upload this as a separate file labeled 'Manuscript'.

We look forward to receiving your revised manuscript.

Kind regards,

Shahabedin Rahmatizadeh, Ph.D.

Academic Editor

PLOS ONE

Journal Requirements:

2. In your Methods section, please include additional information about your dataset and ensure that you have included a statement specifying whether the collection and analysis method complied with the terms and conditions for the source of the data.

3. In the online submission form, you indicated that [All relevant data is within the manuscript and its Supporting Information files upon reasonable request from the corresponding author]. 

4. We note that your Data Availability Statement is currently as follows: [All relevant data is within the manuscript and its Supporting Information files upon reasonable request from the corresponding author]. 

Additional Editor Comments:

The paper is well prepared, but it is necessary to address the issues raised by the original reviewers and to make major revisions to the paper.

Some of the statistics provided are out of date and not reliable at this time.

It may be better to focus on broader issues in this area instead of limiting the discussion to one oral and dental problem. A wider range of web-based information should be explored to reach a larger user base.

Please provide a list of some reputable and useful websites, for example, top ten websites.

Make suggestions for improving the status of websites and discuss the role of health practitioners, managers and policy makers.

Reviewers' comments:

Reviewer's Responses to Questions

**Comments to the Author**

1. Is the manuscript technically sound, and do the data support the conclusions?

Reviewer #1: Partly

Reviewer #2: Yes

Reviewer #3: Yes

2. Has the statistical analysis been performed appropriately and rigorously? 

Reviewer #1: Yes

Reviewer #2: Yes

Reviewer #3: Yes

3. Have the authors made all data underlying the findings in their manuscript fully available?

Reviewer #1: Yes

Reviewer #2: Yes

Reviewer #3: Yes

4. Is the manuscript presented in an intelligible fashion and written in standard English?

Reviewer #1: Yes

Reviewer #2: Yes

Reviewer #3: Yes

5. Review Comments to the Author

Reviewer #1: Investigating how patients seek and incorporate information about oral health is vitally important for understanding the questions they ask dental professionals and the types of treatments they prefer. You have done a tremendous amount of work on this study, and I appreciate the time and care you spent on the analysis.

I do, however, have a few questions about this manuscript.

In the first paragraph of the Introduction, you cite a paper from the WHO which states 71% of internet users seek health-related information. A few pages later, you cite a paper by Cline and Haynes which states that there are over 70,000 websites that provide health related content that are visited by more than 50 million users. These numbers seemed a bit off to me, and a quick check of the References showed that these papers are from 2007 and 2001 respectively. When reporting on the information seeking habits of internet users staying completely current is difficult as tools literally change daily, but there must be better sources of data than these. A quick check of Google Scholar (which admittedly is not without its own concerns) shows that Cline and Haynes have been cited more than 2,300 times with more than 500 of these citations coming in the last five years. Perhaps one of these papers would have more recent data that could be used instead.

Too much of the Introduction is also devoted to the various causes of hypersensitivity and tooth pain. This would be relevant if patients were searching for specific causes of their discomfort (such as something like “tooth sensitivity cold” or “tooth sensitivity hard toothbrush”), but the searches documented only show "Sensitive teeth" and "Teeth hypersensitivity" which are much broader.

I also have a question about this question itself; you write, “In the field of dentistry, patients often seek health information regarding teeth hypersensitivity. It is common for patients to turn to the internet for information regarding dental health issues such as tooth hypersensitivity”. Is this anecdotal? Is this a topic patients often ask about in the office? I did a quick search in Google Trends for the broader term you used (Tooth Sensitivity) and it didn’t register as a trending topic. I broadened this out to the last year and compared it to what Google reported as a more common related search (Tooth Pain) and here are the results:

(See attachment for image)

I don’t have access to more powerful analytics platforms like Semrush or Similarweb, but I did a similar search using Wikipedia’s Pageviews Analysis for Tooth Sensitivity and Toothache (which is the parent term for Tooth Sensitivity on the platform):

(See attachment for image)

I have no doubt that Tooth Sensitivity is of supreme interest to patients suffering from its effects, but it’s not as common as other concerns to internet users. Was this topic chosen to make analyzing the results more manageable? If so this is fine, real world concerns like this are valid, but this should be noted if this indeed a conscious choice.

In the Methods and Materials you mention that you used Google Chrome in Incognito mode to minimize impact from cookies and to eliminate bias. As you may have read, Google recently settled a class-action lawsuit in the United States due to years of misleading claims about the degree of anonymity provided by incognito mode. I realize that this information was made public after this manuscript was prepared and after your study was conducted, but the objectivity of Google’s SEO has always been suspect. I appreciate the steps you took to try to make these searches as transparent and reproducible as possible, but using a search engine like DuckDuckGo or a browser like Tor would be a better option.

While thinking about the Methods, how did you decide upon analyzing a total of 600 websites? With over 30,000,000 results choosing a Representative Sample (say even 1%) would be daunting, but there needs to be some explanation as to how this choice was made. Did you consult with a statistician? Did you follow the approach used by a similar study? Did you try to mimic the behavior of the average searcher? If this is the case 100 results from each search is far too large as previous studies have noted that the vast majority of users rarely click through more than the second page of results, so perhaps the first 20 would be a more common sample size. It’s also true that users can rarely access more than the first 1,000 results from a search engine, so creating a true Representative Sample is difficult without access to advanced tools and metrics.

If you were to employ Systematic Sampling you could use each search engine’s Advanced Search features to categorize the results. For example, in Google’s Advanced Search you could limit by Site or Domain. You stratified the websites selected for analysis by dental or medical centers, commercial entities, non-profits and governments or universities. You could do this in Google by choosing domains like .com, .org, .gov and .edu. This would allow you to choose the top 100 (or whatever number you choose) websites from each domain. This may lead to a fairer comparison of the readability and quality of offerings from each type of entity.

Reviewer #2: The article provides a rigorous study of the websites in question using reasonable evaluation methodologies. The data is available, with the caveat that the authors say "All relevant data is within the manuscript and its Supporting Information files upon reasonable request from the corresponding author". According to the PLOS ONE data availability standards the supporting information files must be provided to PLOS ONE at the time of publication so that they can be added their Figshare Repository or uploaded to a public data repository.

While the article is strong, there are some areas that should be revised. In the Methods area the authors mention utilizing four search engines, while later only utilizing three. The Methods are largely repeated in the Discussion section, in a manner which is superfluous. On page 12 the authors note they intend to address how reader evaluate the data on the sites, although this is only addressed tangentially in the paper. Most of the evaluation information seems to be focused on professional evaluation. That reference should be clarified or removed unless the authors choose to add a substantive section addressing that point.

On page 16, where the authors present their keywords, they have placed them in parantheses. This has a special significance with those search engines (phrase searching) and may be confusing to some readers since I do not believe from the text that these were searched as a phrase. If they were this should be specifically pointed out. If they were not they should be emphasized with some other mechanism, such as italics.

On page 18, in the sentences "Since sensitivity is not usually an emergency situation,

people are more likely to search for online health information to rule out negative outcomes or

consult a dentist [30]. This preferred choice is based on people's tendency to accept positive

information about themselves, such as suggestions of good health, rather than negative

information derived from themselves, such as indications of illness [31]. This can be of huge

importance as patients are likely to take a more proactive role in managing their own healthcare

[32]." the authors make a number of broad assertions that are only supported by single sources per assertion. They should moderate that language or supply more substantive support for those statements.

There seems to be text missing at the beginning of page nineteen, resulting in an incomplete sentance at the beginning of the page.

Some additional items should be added to the limitations section. While three search engines are used for the evaluation, two of them (bing and yahoo) share the same indexing and much of the same technical back end. This results in similar, but not identical, search results. It should also be noted that Incognito mode does not block IP address information from being communicated to the search tool, meaning that search results are still influenced by data deriveable from IP address, such as geographic location.

In my judgement, when these revisions are made and the authors perform the appropriate data sharing steps this article will be suitable for publication.

Reviewer #3: The Introduction section needs clearer transitions, particularly: in the beginning where the authors describe information seeking behavior then abruptly change to describing tooth sensitivity and its potential causes; the middle section about pain tolerance (which doesn't seem particularly relevant to the manuscript); and then toward the end when the authors return to information seeking behavior. Adding paragraphs to the Introduction when a new concept its introduced would also be helpful.

There is a great deal of repetition in the Introduction and Discussion sections. I would edit down the repeated ideas to one or the other sections.

The conclusion does not include a recommendation for additional research in this area, which is a point the authors seem to be making in other areas of the manuscript. I suggest including explicit language that describes the weaknesses in the currently available literature on tooth sensitivity and how those may be strengthened.

The data and statistical analyses appear to be valid and are sound. It may be helpful to include a list of the 90 sites that met the review criteria.

6. PLOS authors have the option to publish the peer review history of their article (what does this mean?). If published, this will include your full peer review and any attached files.

Reviewer #1: No

Reviewer #2: No

Reviewer #3: No

---

## [Author Response · Author response to Decision Letter 0]

4 Jul 2024

Journal Requirements:

2. In your Methods section, please include additional information about your dataset and ensure that you have included a statement specifying whether the collection and analysis method complied with the terms and conditions for the source of the data.

Response: We added the following statement to method section “Our dataset was sourced from publicly available websites, with data collection conducted through manually searching the search engines and collecting the data in a spreadsheet, and analysis performed using SPSS; all processes strictly adhered to the terms and conditions specified by the data provider”.

3. In the online submission form, you indicated that [All relevant data is within the manuscript and its Supporting Information files upon reasonable request from the corresponding author]. 

4. We note that your Data Availability Statement is currently as follows: [All relevant data is within the manuscript and its Supporting Information files upon reasonable request from the corresponding author]. 

Response: We apologize for any inconvenience caused. We would like to confirm that all relevant data is included within the manuscript. Please accept our sincerest apologies.

Additional Editor Comments:

The paper is well prepared, but it is necessary to address the issues raised by the original reviewers and to make major revisions to the paper.

Some of the statistics provided are out of date and not reliable at this time.

Response: We have updated the references for statistics in our manuscript to reflect the most current data available.

It may be better to focus on broader issues in this area instead of limiting the discussion to one oral and dental problem. A wider range of web-based information should be explored to reach a larger user base.

Response: We acknowledge the suggestion to broaden the scope of our discussion. However, the focus of this study is specifically on dental hypersensitivity due to the unique nature of this condition and its impact on patients. While there is extensive research available on various other oral health conditions such as oral malignancies, caries, and periodontitis, there is a relative paucity of studies that exclusively examine online information related to dental hypersensitivity. This study aims to fill this gap by providing a detailed assessment of the quality and readability of online health information specific to dental hypersensitivity. Our findings contribute to a more understanding of how this condition is represented on the internet, which is valuable for both patients and healthcare providers.

Please provide a list of some reputable and useful websites, for example, top ten websites.

Response: To address the request for a list of reputable and useful websites, we have included a comprehensive list of the top ten websites in Appendix 1.

Make suggestions for improving the status of websites and discuss the role of health practitioners, managers and policy makers.

Response: we have added a section to the discussion including recommendations to improve the quality of online information and the role of health practitioners, managers and policy makers.

Reviewer #1 comments:

 Investigating how patients seek and incorporate information about oral health is vitally important for understanding the questions they ask dental professionals and the types of treatments they prefer. You have done a tremendous amount of work on this study, and I appreciate the time and care you spent on the analysis.

I do, however, have a few questions about this manuscript.

In the first paragraph of the Introduction, you cite a paper from the WHO which states 71% of internet users seek health-related information. A few pages later, you cite a paper by Cline and Haynes which states that there are over 70,000 websites that provide health related content that are visited by more than 50 million users. These numbers seemed a bit off to me, and a quick check of the References showed that these papers are from 2007 and 2001 respectively. When reporting on the information seeking habits of internet users staying completely current is difficult as tools literally change daily, but there must be better sources of data than these. A quick check of Google Scholar (which admittedly is not without its own concerns) shows that Cline and Haynes have been cited more than 2,300 times with more than 500 of these citations coming in the last five years. Perhaps one of these papers would have more recent data that could be used instead.

Response: Thank you for bringing this to our attention. We have made the necessary updates to the information, which now includes a new reference published in 2022.

“Due to the widespread availability of smartphones, tablets, and laptops, a considerable percentage of individuals, estimated at approximately 52%, have resorted to utilizing the internet as a source for health-related information and symptoms in 2022. Notably, this upward trend in dependence on digital platforms for health-related information is expected to endure [20].”

Too much of the Introduction is also devoted to the various causes of hypersensitivity and tooth pain. This would be relevant if patients were searching for specific causes of their discomfort (such as something like “tooth sensitivity cold” or “tooth sensitivity hard toothbrush”), but the searches documented only show "Sensitive teeth" and "Teeth hypersensitivity" which are much broader.

 Response: Thank you for your valuable comment. The focus on various causes of hypersensitivity and tooth pain in the Introduction was intended to provide a comprehensive background on the topic. While the searches documented in our study were broad ("Sensitive teeth" and "Teeth hypersensitivity"), we aimed to contextualize these terms within the broader spectrum of dental health issues that individuals may encounter. Understanding the potential causes of tooth sensitivity can help frame the significance of these search behaviors and provide a foundation for interpreting the findings. Moreover, discussing these causes allows us to highlight the complexity of dental hypersensitivity and its impact on individuals seeking information online. Even though the specific search terms captured in our study were general, the underlying causes we discussed can influence how patients perceive and search for information related to their dental discomfort.

We believe that providing this background enhances the reader's understanding of the motivations behind online searches related to tooth sensitivity. It underscores the importance of addressing both specific and general queries in dental health information-seeking behaviors. We have taken your feedback into consideration and have revised the Introduction section to ensure a balanced presentation of background information while maintaining relevance to our study's focus. We hope these revisions better align with your expectations.

I also have a question about this question itself; you write, “In the field of dentistry, patients often seek health information regarding teeth hypersensitivity. It is common for patients to turn to the internet for information regarding dental health issues such as tooth hypersensitivity”. Is this anecdotal? Is this a topic patients often ask about in the office? I did a quick search in Google Trends for the broader term you used (Tooth Sensitivity) and it didn’t register as a trending topic. I broadened this out to the last year and compared it to what Google reported as a more common related search (Tooth Pain) and here are the results:

I don’t have access to more powerful analytics platforms like Semrush or Similarweb, but I did a similar search using Wikipedia’s Pageviews

Analysis for Tooth Sensitivity and Toothache (which is the parent term for Tooth Sensitivity on the platform):

I have no doubt that Tooth Sensitivity is of supreme interest to patients suffering from its effects, but it’s not as common as other concerns to internet users. Was this topic chosen to make analyzing the results more manageable? If so this is fine, real world concerns like this are valid, but this should be noted if this indeed a conscious choice.

Response: Dental hypersensitivity, while not always trending as highly as broader terms like "tooth pain," is a significant concern for those affected. In clinical practice, patients frequently inquire about this condition, seeking advice on management and treatment options. In addition, “tooth pain” as a term can include many conditions that might have good or poor quality of information in the web, however, this will not answer the question about the quality of online information about the dental hypersensitivity. In addition, the analysis of Wikipedia pageviews showing higher interest in "toothache" compared to "tooth sensitivity" aligns with the broader search trends observed. However, this study's focus on dental hypersensitivity is intentional and aims to shed light on a condition that, while perhaps less frequently searched, significantly impacts those who experience it. By concentrating on dental hypersensitivity, we can provide a thorough evaluation of the quality of online information and identify areas for improvement, ultimately benefiting patients and healthcare providers.

In the Methods and Materials you mention that you used Google Chrome in Incognito mode to minimize impact from cookies and to eliminate bias. As you may have read, Google recently settled a class-action lawsuit in the United States due to years of misleading claims about the degree of anonymity provided by incognito mode. I realize that this information was made public after this manuscript was prepared and after your study was conducted, but the objectivity of Google’s SEO has always been suspect. I appreciate the steps you took to try to make these searches as transparent and reproducible as possible, but using a search engine like DuckDuckGo or a browser like Tor would be a better option.

Response: Thank you for your insightful comment regarding our use of Google Chrome in Incognito mode in our Methods and Materials section. We appreciate your concern regarding the recent developments related to Google's Incognito mode and its implications for SEO objectivity.

We acknowledge that the information regarding Google's settlement in the United States came to light after our manuscript was prepared and our study conducted. Our decision to use Google Chrome in Incognito mode was based on the aim to minimize the impact of personalized search results and cookies, thereby enhancing the reproducibility and transparency of our search queries.

To address your valid suggestion, we agree that exploring alternative search engines like DuckDuckGo or employing browsers such as Tor could offer additional insights into how search results vary across different platforms known for prioritizing user privacy. This consideration could indeed contribute to a more comprehensive understanding of how search engine biases may influence the findings of studies similar to ours.

In our limitations section, we expanded upon this topic by discussing the implications of using different search engines and browsing modes on the objectivity and reproducibility of online search studies. This addition will highlight the importance of considering alternative methods to further enhance the reliability of future research in this area.

While thinking about the Methods, how did you decide upon analyzing a total of 600 websites? With over 30,000,000 results choosing a Representative Sample (say even 1%) would be daunting, but there needs to be some explanation as to how this choice was made. Did you consult with a statistician? Did you follow the approach used by a similar study? Did you try to mimic the behavior of the average searcher? If this is the case 100 results from each search is far too large as previous studies have noted that the vast majority of users rarely click through more than the second page of results, so perhaps the first 20 would be a more common sample size. It’s also true that users can rarely access more than the first 1,000 results from a search engine, so creating a true Representative Sample is difficult without access to advanced tools andmetrics.

Response: We followed an approach similar to previous studies evaluating online health information. These studies often use a predefined number of websites to ensure manageability and representativeness. Consulting relevant literature helped us determine that analyzing 600 websites would provide a robust dataset. While it is acknowledged that most users rarely click beyond the second page of search results, analyzing the first 100 results from each search engine allowed us to capture a broader range of information, ensuring inclusion of less prominent but relevant websites. This approach helps us understand the quality of information available to users who might explore beyond the initial few results.

If you were to employ Systematic Sampling you could use each search engine’s Advanced Search features to categorize the results. For example, in Google’s Advanced Search you could limit by Site or Domain. You stratified the websites selected for analysis by dental or medical centers, commercial entities, non-profits and governments or universities. You could do this in Google by choosing domains like .com, .org, .gov and .edu. This would allow you to choose the top 100 (or whatever number you choose) websites from each domain. This may lead to a fairer comparison of the readability and quality of offerings from each type of entity.

Response: Your suggestion to employ systematic sampling and use search engine’s advanced features is valuable. For this study, we manually categorized websites based on their affiliation (dental or medical centers, commercial entities, non-profits, and governments or universities). Future research could indeed benefit from leveraging advanced search features to further stratify and select websites, ensuring an even more balanced comparison of readability and quality across different types of entities. While users can rarely access more than the first 1,000 results from a search engine, our method aimed to balance the depth of information with practical limitations. By stratifying the websites and ensuring diverse representation, we aimed to provide a fair and comprehensive analysis.

Reviewer #2 comments: 

The article provides a rigorous study of the websites in question using reasonable evaluation methodologies. 

The data is available, with the caveat that the authors say "All relevant data is within the manuscript and its Supporting Information files upon reasonable request from the corresponding author". According to the PLOS ONE data availability standards the supporting information files must be provided to PLOS ONE at the time of publication so that they can be added their Figshare Repository or uploaded to a public data repository.

Response: We will ensure that all relevant data is made available in accordance with PLOS ONE's data availability standards. Specifically, we will include all supporting information files in Appendix

---

## [Decision Letter · Decision Letter 1]

19 Aug 2024

PONE-D-24-10064R1An Infodemiologic Review of Internet Resources on Dental Hypersensitivity: A Quality and Readability Assessment.PLOS ONE

Dear Dr. Alassaf,

Thank you for submitting your manuscript to PLOS ONE. After careful consideration, we feel that it has merit but does not fully meet PLOS ONE’s publication criteria as it currently stands. Therefore, we invite you to submit a revised version of the manuscript that addresses the points raised during the review process.

We look forward to receiving your revised manuscript.

Kind regards,

Shahabedin Rahmatizadeh, Ph.D.

Academic Editor

PLOS ONE

Journal Requirements:

Additional Editor Comments:

Most of the points requested by my reviewers have been addressed, and only one issue remains. Please submit the revised manuscript after making the necessary corrections.

Reviewers' comments:

Reviewer's Responses to Questions

**Comments to the Author**

1. If the authors have adequately addressed your comments raised in a previous round of review and you feel that this manuscript is now acceptable for publication, you may indicate that here to bypass the “Comments to the Author” section, enter your conflict of interest statement in the “Confidential to Editor” section, and submit your "Accept" recommendation.

Reviewer #2: All comments have been addressed

Reviewer #3: (No Response)

2. Is the manuscript technically sound, and do the data support the conclusions?

Reviewer #2: Yes

Reviewer #3: Yes

3. Has the statistical analysis been performed appropriately and rigorously? 

Reviewer #2: Yes

Reviewer #3: I Don't Know

4. Have the authors made all data underlying the findings in their manuscript fully available?

Reviewer #2: Yes

Reviewer #3: Yes

5. Is the manuscript presented in an intelligible fashion and written in standard English?

Reviewer #2: Yes

Reviewer #3: Yes

6. Review Comments to the Author

Reviewer #2: I feel the author has adequately addressed my feedback and I feel the paper is suitable for publication.

Reviewer #3: Minor error in Figure 2 - "Fair Eaesy" is mispelled. Will need to be corrected in the software that produced the figure.

7. PLOS authors have the option to publish the peer review history of their article (what does this mean?). If published, this will include your full peer review and any attached files.

Reviewer #2: No

Reviewer #3: No

---

## [Author Response · Author response to Decision Letter 1]

22 Aug 2024

Dear respected Editor and Reviewers,

Spelling mistake corrected in the figure 

Thank You

---

## [Decision Letter · Decision Letter 2]

4 Oct 2024

PONE-D-24-10064R2An Infodemiologic Review of Internet Resources on Dental Hypersensitivity: A Quality and Readability Assessment.PLOS ONE

Dear Dr. Alassaf,

Thank you for submitting your manuscript to PLOS ONE. After careful consideration, we feel that it has merit but does not fully meet PLOS ONE’s publication criteria as it currently stands. Therefore, we invite you to submit a revised version of the manuscript that addresses the points raised during the review process.

We look forward to receiving your revised manuscript.

Kind regards,

Shahabedin Rahmatizadeh, Ph.D.

Academic Editor

PLOS ONE

Journal Requirements:

Additional Editor Comments:

Thank you for revising the article. Considering the reviewers' comments, please address the mentioned issues and submit the revised version to the journal.

Reviewers' comments:

Reviewer's Responses to Questions

**Comments to the Author**

1. If the authors have adequately addressed your comments raised in a previous round of review and you feel that this manuscript is now acceptable for publication, you may indicate that here to bypass the “Comments to the Author” section, enter your conflict of interest statement in the “Confidential to Editor” section, and submit your "Accept" recommendation.

Reviewer #4: All comments have been addressed

Reviewer #5: All comments have been addressed

Reviewer #6: (No Response)

Reviewer #7: All comments have been addressed

2. Is the manuscript technically sound, and do the data support the conclusions?

Reviewer #4: Yes

Reviewer #5: Partly

Reviewer #6: Yes

Reviewer #7: Yes

3. Has the statistical analysis been performed appropriately and rigorously? 

Reviewer #4: Yes

Reviewer #5: I Don't Know

Reviewer #6: Yes

Reviewer #7: Yes

4. Have the authors made all data underlying the findings in their manuscript fully available?

Reviewer #4: Yes

Reviewer #5: No

Reviewer #6: Yes

Reviewer #7: Yes

5. Is the manuscript presented in an intelligible fashion and written in standard English?

Reviewer #4: Yes

Reviewer #5: Yes

Reviewer #6: Yes

Reviewer #7: Yes

6. Review Comments to the Author

Reviewer #4: As demonstrated in the final manuscript, the authors have made a significant effort in applying the reviewers' corrections, and therefore, I believe this research deserves to be published.

Reviewer #5: The topic is interesting. However, I would like to recommend the author to-

1. explain why the instuments were chosen to use in this study

2. explain the technical terms specifically for those which are essential to the study such as the Kruskal-Wallis test, as the readers could more understanding.

3. specify in the meterial and method part regarding the websites that used to collect data and explain why they were chosen.

4. In discussion, the keywords "dentin sensitivity, cervical hypersensitivity, cemental

hypersensitivity, and root hypersensitivity" are different from the result in Meterial and Methods part.

5. For future study, it is recommend that the author should use the information sources that donot share the same index. So, this could be beneficial for the readers to choose the searche engine when they want more or different information since Bing and Yahoo share the same indexing.

6. It is stongly recommended that the significances-benefits or outcomes, of this study should be clearly highlighted in the conclusion.

Reviewer #6: This paper is well written and easy to read.

Font size or a couple of citations in references look inconsistent. This also happens to a couple of pages in the body of text.

Reviewer #7: In general, the author has answered the reviewer(s) ' questions, which improves the quality of the manuscript in this version. Some additional suggestions are listed below.

1. Author affiliation.

Changing an author's affiliation should be done according to the rules of the journal. In general, the unit that recognizes a researcher means that the research was completed during his tenure at that unit, not that it changed when he changed jobs.

2. Typos. The system shows that the author claims to use four search engines, but the article talks about three (i.e., Yahoo, Bing, and Google). Please correct this.

3. Limitation and direction.

The author(s) stated that they followed the literature guidelines regarding the three search engines used (Yahoo, Bing, and Google) and listed their limitations in terms of information coverage in the limitations section of the study.

In fact, due to Internet use regulations and other restrictions, some users in China may also use Baidu to search for English-language health information. Perhaps the researchers can list future research directions, such as comparing the results of search engines with different user bases.

7. PLOS authors have the option to publish the peer review history of their article (what does this mean?). If published, this will include your full peer review and any attached files.

Reviewer #4: No

Reviewer #5: No

Reviewer #6: No

Reviewer #7: **Yes: **Kai-Yu Tang

---

## [Author Response · Author response to Decision Letter 2]

5 Oct 2024

Dear respected editor and reviewers,

We would like to extend our thanks to the editor and the respected reviewers for the valuable and constructive comments. We confirm that we have gone over all comments, responded to all comments/suggestions and done all required corrections, as outlined below. All corrections/additions in the text have been marked in blue. 

Journal Requirements: 

Author’s Response: Thank you for your comment. Upon careful review, we confirm that no changes were made to the reference list in the revised manuscript, and no retracted articles were cited. The reference list remains complete and correct as per the original submission. 

Additional Editor Comments: 

Thank you for revising the article. Considering the reviewers' comments, please address the mentioned issues and submit the revised version to the journal. 

Author’s Response: Thank you for your feedback. We appreciate the opportunity to revise our article based on the reviewers' valuable comments. We have carefully addressed all the issues raised and made the necessary revisions in the manuscript. Thank you for your thoughtful feedback. 

Reviewer #4: As demonstrated in the final manuscript, the authors have made a significant effort in applying the reviewers' corrections, and therefore, I believe this research deserves to be published.

Author’s Response: We truly appreciate your recognition of the effort put into addressing the reviewers' corrections. We are pleased that the revisions have met the expectations and standards of the review process, and we are grateful for your support in recommending this research for publication.

Reviewer #5: The topic is interesting. However, I would like to recommend the author to-

1. explain why the instruments were chosen to use in this study 

Author’s Response: Thank you for your comment. We mentioned the important role of each tool in method section and based on your comment we added the following to discussion section “The HONcode, DISCERN, and JAMA Benchmarks are among the most widely utilized instruments for evaluating the reliability and quality of information presented on medical websites” 

2. explain the technical terms specifically for those which are essential to the study such as the Kruskal-Wallis test, as the readers could more understanding.

Author’s Response: Thank you for your comment. In response to your feedback, we have amended the statistical analysis subsection to include the following: "The Kruskal-Wallis test is a non-parametric statistical test utilized to assess whether there are significant differences between the medians of three or more independent groups. This test serves as an alternative to the one-way ANOVA when the assumptions of normality are not satisfied."

3. specify in the meterial and method part regarding the websites that used to collect data and explain why they were chosen.

Author’s Response: Thank you for your comment. Part to explain as mentioned “Using Google Chrome, version 81.0.4044, we conducted searches on Yahoo! (http://www.yahoo.com), Bing (http://www.bing.com), and Google (http://www.google.com). This approach is in line with the findings of "The Pew Research Center's Internet and American Life Project" [2], which reports that 79% of individuals who search for health information online utilize these search engines"

4. In discussion, the keywords "dentin sensitivity, cervical hypersensitivity, cemental hypersensitivity, and root hypersensitivity" are different from the result in Meterial and Methods part.

Author’s Response: Thank you for directing our attention to this matter; we have amended the inconsistency. 

5. For future study, it is recommend that the author should use the information sources that donot share the same index. So, this could be beneficial for the readers to choose the searche engine when they want more or different information since Bing and Yahoo share the same indexing.

Author’s Response: Thank you for this valuable suggestion we added a brief paragraph based on your suggestion to discussion. In future studies, diversifying information sources beyond those sharing the same index, such as Bing and Yahoo, can enhance the breadth of research findings. By incorporating varying search engines, readers gain access to a wider array of perspectives and data, allowing for more comprehensive and varied insights. This approach not only enriches the research but also aids readers in selecting search engines that better meet their informational needs.

6. It is strongly recommended that the significances-benefits or outcomes, of this study should be clearly highlighted in the conclusion.

Author’s Response: Thank you for this valuable suggestion we added a brief statement based on your suggestion to conclusion. “This study highlights the moderate quality and challenging readability of online health information on dental sensitivity. The findings underscore the need for improved reference standards and HONcode certification to enhance reliability. By identifying gaps in current online resources, this research emphasizes the importance of developing accessible and trustworthy health information for patients.”

Reviewer #6: 

This paper is well written and easy to read.

Font size or a couple of citations in references look inconsistent. This also happens to a couple of pages in the body of text.

Author’s Response: Amended

Reviewer #7: 

In general, the author has answered the reviewer(s) ' questions, which improves the quality of the manuscript in this version. 

Author’s Response: Thank you for your positive feedback. We are glad to hear that the revisions have satisfactorily addressed the reviewers' questions and have contributed to improving the quality of the manuscript.

Some additional suggestions are listed below.

1. Author affiliation.

Changing an author's affiliation should be done according to the rules of the journal. In general, the unit that recognizes a researcher means that the research was completed during his tenure at that unit, not that it changed when he changed 

jobs.

Author’s Response: 

I would like to kindly inform you that the research presented in this manuscript was conducted during my tenure at my current affiliation, King Fahad Hospital. At the time of conducting this study, I was concurrently affiliated with Taibah University, but all the research work and contributions were completed while I was affiliated with King Fahad Hospital.

I believe it is appropriate to reflect my current affiliation in the publication, and I appreciate your understanding in this matter. Please let me know if you need any additional information or clarification.

2. Typos. The system shows that the author claims to use four search engines, but the article talks about three (i.e., Yahoo, Bing, and Google). Please correct this.

Author’s Response: Thank you for directing our attention to this matter; we have amended the inconsistency. 

3. Limitation and direction.

The author(s) stated that they followed the literature guidelines regarding the three search engines used (Yahoo, Bing, and Google) and listed their limitations in terms of information coverage in the limitations section of the study.

In fact, due to Internet use regulations and other restrictions, some users in China may also use Baidu to search for English-language health information. Perhaps the researchers can list future research directions, such as comparing the results of search engines with different user bases.

Thank you for this valuable suggestion we added a brief paragraph based on your suggestion to discussion. “Future research could explore the effectiveness and coverage of other search engines like Baidu, especially in regions with different internet regulations. By comparing search engines with distinct user bases, such as Baidu, Yahoo, Bing, and Google, researchers can gain insights into how users access English-language health information across diverse geographic and cultural contexts. This could help identify gaps in information availability and inform strategies to improve online health information dissemination globally. Additionally, examining user behavior and preferences on these platforms could provide a deeper understanding of search engine optimization and its impact on health literacy.”

---

## [Decision Letter · Decision Letter 3]

15 Oct 2024

An Infodemiologic Review of Internet Resources on Dental Hypersensitivity: A Quality and Readability Assessment.

PONE-D-24-10064R3

Dear Dr. Muath Saad Alassaf,

We’re pleased to inform you that your manuscript has been judged scientifically suitable for publication and will be formally accepted for publication once it meets all outstanding technical requirements.

Kind regards,

Shahabedin Rahmatizadeh, Ph.D.

Academic Editor

PLOS ONE

Additional Editor Comments (optional):

I would like to express my gratitude for your attention to the points raised by the journal reviewers and for addressing the requested revisions. The article is deemed to meet the requisite standards for acceptance.

Reviewers' comments:

Reviewer's Responses to Questions

**Comments to the Author**

1. If the authors have adequately addressed your comments raised in a previous round of review and you feel that this manuscript is now acceptable for publication, you may indicate that here to bypass the “Comments to the Author” section, enter your conflict of interest statement in the “Confidential to Editor” section, and submit your "Accept" recommendation.

Reviewer #6: All comments have been addressed

Reviewer #7: All comments have been addressed

2. Is the manuscript technically sound, and do the data support the conclusions?

Reviewer #6: Yes

Reviewer #7: Partly

3. Has the statistical analysis been performed appropriately and rigorously? 

Reviewer #6: Yes

Reviewer #7: Yes

4. Have the authors made all data underlying the findings in their manuscript fully available?

Reviewer #6: Yes

Reviewer #7: (No Response)

5. Is the manuscript presented in an intelligible fashion and written in standard English?

Reviewer #6: Yes

Reviewer #7: Yes

6. Review Comments to the Author

Reviewer #6: (No Response)

Reviewer #7: The authors have answered the reviewers' questions in this version. This improves the quality of the manuscript.

7. PLOS authors have the option to publish the peer review history of their article (what does this mean?). If published, this will include your full peer review and any attached files.

Reviewer #6: No

Reviewer #7: **Yes: **Kai-Yu Tang

---

## [Editor Report · Acceptance letter]

12 Nov 2024

PONE-D-24-10064R3 

PLOS ONE

Dear Dr. Alassaf, 

I'm pleased to inform you that your manuscript has been deemed suitable for publication in PLOS ONE. Congratulations! Your manuscript is now being handed over to our production team.

Kind regards, 

on behalf of

Dr. Shahabedin Rahmatizadeh 

Academic Editor

PLOS ONE